# A Metabolomic Study of the Variability of the Chemical Composition of Commonly Consumed Coffee Brews

**DOI:** 10.3390/metabo9010017

**Published:** 2019-01-18

**Authors:** Joseph A. Rothwell, Erikka Loftfield, Roland Wedekind, Neal Freedman, Callie Kambanis, Augustin Scalbert, Rashmi Sinha

**Affiliations:** 1International Agency for Research on Cancer (IARC), Nutrition and Metabolism Section, Biomarkers Group, 150 Cours Albert Thomas, CEDEX 08, F-69372 Lyon, France; RothwellJ@fellows.iarc.fr (J.A.R.); WedekindR@students.iarc.fr (R.W.); 2National Cancer Institute, 9609 Medical Center Drive, Bethesda, MD 20892, USA; erikka.loftfield@nih.gov (E.L.); freedmanne@mail.nih.gov (N.F.); calkam15@terpmail.umd.edu (C.K.); sinhar@exchange.nih.gov (R.S.)

**Keywords:** coffee, untargeted metabolomics, chemical composition, brew method, roast, bean variety

## Abstract

Coffee drinking has been associated with a lower risk of certain chronic diseases and overall mortality. Its effects on disease risk may vary according to the type of coffee brew consumed and its chemical composition. We characterized variations in the chemical profiles of 76 coffee brew samples representing different brew methods, roast levels, bean species, and caffeine types, either prepared or purchased from outlets in Rockville, Maryland, United States of America. Samples were profiled using liquid chromatography coupled with high-resolution mass spectrometry, and the main sources of chemical variability identified by the principal component partial R-square multivariable regression were found to be brew methods (R_partial_^2^ = 36%). A principal component analysis (PCA) was run on 18 identified coffee compounds after normalization for total signal intensity. The three first principal components were driven by roasting intensity (41% variance), type of coffee beans (29%), and caffeine (8%). These variations were mainly explained by hydroxycinnamoyl esters and diketopiperazines (roasting), N-caffeoyltryptophan, N-*p-*coumaroyltryptophan, feruloylquinic acids, and theophylline (coffee bean variety) and theobromine (decaffeination). Instant coffees differed from all coffee brews by high contents of diketopiperazines, suggesting a higher roast of the extracted beans. These variations will be important to consider for understanding the effects of different coffee brews on disease risk.

## 1. Introduction

Coffee drinking has been consistently linked to lower risk of overall mortality [1,2,3,4], diabetes [5], cardiovascular diseases [6], and cancers of the liver [7,8] and endometrium [9,10] in epidemiological studies. Possible harmful effects have also been examined, but no consistent evidence of harmful associations with diseases could be found, except for outcomes related to pregnancy and the risk of fractures in women [11]. The mechanisms underlying these associations are likely mediated by any of the thousands of specific coffee phytochemicals present in the beans or produced during roasting. Coffee is prepared in many diverse ways, such as adding water to instant coffee powder, percolation, or filtering of ground coffee, and preparation of espresso in commercial machines. In addition, raw coffee beans include both Arabica and Robusta species that are then roasted to different degrees. All these parameters affect the levels of bioactive compounds in the coffee [12,13,14] and may influence the effects of coffee consumption on disease risk.

Epidemiological studies on coffee and disease risk require accurate measurements of coffee intake. However, food frequency questionnaires (FFQ) commonly used to assess coffee intake almost always lack information on coffee type and preparation. As such, these factors are rarely considered in epidemiological studies, and it remains unclear whether they affect associations with health. Investigating the composition of popular coffee brews and determining which parameters have the greatest influence on the content of important coffee bioactives should assist in the creation of a focused coffee FFQ module and food composition table. It will also inform the interpretation of coffee biomarkers measured in epidemiological studies [15,16,17].

Some previous studies on coffee composition have analyzed a wide range of brewed coffees, but they have typically focused on specific components of interest, such as flavor volatiles, caffeine, or lipids rather than overall profiles [18,19,20,21]. The aim of this study was to analyze through an agnostic untargeted metabolomic approach a wide range of coffee brews in order to identify coffee characteristics that have the greatest influence on their chemical profile. 

## 2. Results

### 2.1. Untargeted Constituent Profiles and Variation Due to Different Coffee Parameters

A total of 3670 spectral features were measured across the 76 coffee brews. PC-PR2 analysis revealed that brew method explained the largest proportion of variability in metabolomic data (R_partial_^2^ = 36%), followed by roast (R_partial_^2^ = 16%), bean type (R_partial_^2^ = 9%) and caffeine (R_partial_^2^ = 7%). In relation to technical variables, injection plate and order of injection explained R_partial_^2^ = 10% and 3% of the variability, respectively. The total variability explained by all selected variables (R_model_^2^) was 61% (Appendix A).

The summed signal intensities from all spectral features measured varied with the brew method (Appendix A). On average, espresso and boiled brews produced significantly greater total signal intensity than instant coffee brews, which were taken as a reference for brew strength (mean fold change = 1.62; Student’s *t*-test *p* = 0.0007).

### 2.2. Annotation of Coffee Compounds in Coffee Brew Metabolomic Data

An in-house database on known coffee compounds was built and a targeted list of 64 tentative annotations was established based on the matching of their accurate masses with those of the known coffee compounds (Appendix A). A tentative annotation of 24 compounds was invalidated on the basis of their mass fragmentation spectra and/or retention time (‘unknowns’). The identity of an additional 22 peaks could not be confirmed because of the lack of chemical standards or good quality mass fragmentation spectra (confidence level 4). In contrast, 14 compounds could be identified with a high level of confidence (levels 1 and 2) by comparison of their retention time and MS/MS fragmentation spectra with those of the commercially available reference compounds, or by comparison of their MS/MS spectra with those in public database when standards were not available (Table 1). Four additional compounds were tentatively identified based on their accurate masses, known occurrence in coffee brews, and the presence of fragments characteristic of specific structural residues in the molecule (e.g., coumaroyl group or amino acid group) (confidence level 3).

In total, 18 compounds could be identified or tentatively identified (Table 1). These compounds are mono- and dicaffeoylquinic acids, and similar esters with ferulic acid and *p*-coumaric acid (classically named chlorogenic acids [22]), N-caffeoyltryptophan and N-*p*-coumaroyltryptophan, in which the hydroxycinnamic acid is linked to tryptophan through an amide bond, 3 methylxanthines, cafamarine (a diterpenoid glycoside), 5 diketopiperazines (cyclic dipeptides) and methyl-2-pyrrolecarboxaldehyde.

### 2.3. Association of Coffee Compound Profiles with Different Types of Coffee Brews

Intensity profiles of the 18 coffee compounds in the 76 unique coffee samples were analyzed using principal component analysis (PCA). Data were first normalized over total signal intensities to correct for differences in coffee brew dilution (Appendix A). The first two principal components (PC) accounted for respectively 41% and 29% of the variance in the data. Different brew methods could be separated on the first two PCs. Instant coffee showed lower scores in both PC1 and PC2, and espresso and K-cup coffee brews showed lower scores in PC1 when compared to all other coffee brews (Figure 1A). Loading plots showed that PC1 was largely explained by the high levels of the six phenolic acid esters and cafamarine, and low levels of the 5 diketopiperazines (Figure 1C). The same score plot also showed a clear distinction between coffee brews according to the type of coffee beans used, either Arabica or blends of Arabica and Robusta (Figure 1B). The second PC was mainly explained by high loadings of the two feruloylquinic acid isomers, the two phenolic acid amides, and the five diketopiperazines (Figure 1C). Caffeinated and decaffeinated coffee brews could be differentiated along PC3 (accounting for 7.7% of the variability), a result consistent with the high loadings of paraxanthine and theobromine on this PC (Appendix A).

### 2.4. Coffee Compounds Associated with Coffee Brew Characteristics

*T*-tests were run to identify coffee compounds most strongly associated with roasting, the type of coffee beans, instant coffee, and bean decaffeination (Appendix A), showing significant differences of the relative levels of several coffee compounds after Bonferroni correction. The three caffeoyl esters showed lower concentrations and three diketopiperazines showed higher concentrations in dark roasted brews when compared with medium roasted beans (Figure 2). Light roasted brews could not be compared because of a low number of samples. Coffee brews prepared with Arabica beans showed relatively high concentrations of N-caffeoyltryptophan, N-*p-*coumaroyltryptophan, the two feruloylquinic acid isomers, and theophylline when compared with coffee brews prepared with blends of Arabica and Robusta beans (Figure 3). Instant coffees were also compared to all other coffee brew samples. The most discriminating compounds were the 5 diketopiperazines (Figure 4). Decaffeinated coffees showed lower intensities of theobromine and cafamarine when compared with all caffeinated coffee brews (Figure 5).

## 3. Discussion

Coffee beans contain a wide range of phytochemicals, many of which are potentially bioactive and may explain observed associations with health. These consist of chemical classes as diverse as alkaloids, amino acids, terpenes, lipids, phenolic acids, and many small volatiles that contribute to aroma and bitterness [24]. The aim of this study was to explore the chemical profiles of a wide range of commonly-consumed coffee brews and to determine which coffee characteristics most influence these profiles. Chemical profiles were acquired using high-resolution mass spectrometry and were found to vary substantially according to the type of coffee brew, the extent of roasting of beans, the variety of bean, and decaffeination. The data was mined for known coffee components. Eighteen compounds could be annotated, and PCA analysis was run after normalization over the total solute concentration to correct for variations in dilution. Much of the variations in the composition of these 18 compounds were driven by the type of coffee brew, with instant coffee being clearly different from all other types of coffee under study (Figure 1A).

The analysis of the loadings on the first two PCs provides clues for how to interpret these differences. The first PC is mainly driven by high levels of chlorogenic acids (caffeoyl-, feruloyl-, and *p*-coumaroylquinic acids) and low levels of diketopiperazines (Figure 1C, Figure 2). Diketopiperazines are cyclodipeptides formed by the degradation of proteins contained in green coffee beans during heat treatment [25,26]. They confer a bitter taste to highly roasted coffee. In contrast, chlorogenic acids are largely degraded during roasting [27]. This suggests that coffee brews with low scores on the first PC are more highly roasted than coffee brews with high scores. Instant coffee samples, which show particularly high levels of diketopiperazines (Figure 4), would then differ from other coffee brews by a particularly high degree of roast.

The second PC of the same PCA plot is mainly driven by the variety of coffee beans used to prepare the coffee brews. Coffee brews prepared from Arabica beans show high scores, whereas those prepared from blends of Arabica and Robusta beans show low scores (Figure 1B). The compounds with the highest loadings on the second PC are the two phenolic acid amides (N-caffeoyltryptophan and N-*p*-coumaroyltryptophan) and the two feruloylquinic acid isomers (Figure 1C, Figure 3). This is fully consistent with the known composition of both varieties of beans. The two phenolic acid amides are characteristic of Robusta beans and are either not found or found at very low concentrations in Arabica beans [28,29]. Feruloylquinic acids are known to be particularly abundant in Robusta beans when compared with Arabica beans [29]. The low scores of the instant coffee samples on PC2 are also consistent with the wide use of Robusta beans in the production of instant coffees (https://www.britannica.com/topic/coffee).

The third PC was notably explained by the high negative loadings of theobromine and paraxanthine, two known biosynthetic precursors of caffeine in the coffee plant [30]. T-tests showed in particular relatively low levels of theobromine in decaffeinated coffee samples, as would have been expected (Figure 5). 

The present metabolomic study shows the distinct chemical compositions of the various coffee brew samples. Several classes of coffee compounds (chlorogenic acids, phenolic acid amides, methylxanthines, and diketopiperazines) showed large variations in their concentrations, which were explained by the extent of bean roast, the variety of the coffee bean, and also by decaffeination. These coffee compounds showing wide variations in their concentrations are not only abundant in coffee but also have biological properties which contribute to explain the health effects of coffee. In particular, chlorogenic acids were shown to exhibit a wide range of biological activities, including indirect antioxidant effects through the induction of phase 2 enzyme activities, inhibition of DNA methyltransferase, inhibition of platelet activities, and interference with glucose absorption [24,31,32]. Caffeine, besides its effects as a stimulant of the central nervous system, increases metabolic rate and energy expenditure and has been used as an aid in weight loss, and may contribute to a reduced risk of metabolic syndrome [33]. 

In most epidemiological studies, assessment of coffee intake is generally limited to the number of cups of coffee consumed and sometimes whether the coffee consumed was usually caffeinated or decaffeinated. It will be important in future epidemiological studies on coffee to accurately record the type of coffee consumed in order to understand the impact of various types of coffee brews (brew method, roasting, coffee beans) on disease risk. A previous cross-over study with a dark roast and a light roast coffee brew suggested that the degree of roast may influence coffee health effects [34]. The intake of dark roast coffee was shown to elicit stronger antioxidant effects upon human erythrocytes and to significantly reduce body weight in pre-obese subjects when compared with light roast coffee, which had no effect.

The use of biomarkers may also help to clarify the role of the different types of coffee brews in disease risk in future epidemiological studies. Several biomarkers of habitual coffee intake have been identified in blood [15,35,36,37]. Ratios of diketopiperazines or catechol sulfate over trigonelline measured in serum were recently proposed as indicators of roast of coffee brews consumed in four European countries differing in their coffee drinking habits [17].

The present study also has some limitations. Firstly, the present work is derived from the analysis of data on 18 compounds that could be successfully annotated and analyzed with the analytical method used in this study. Some important compounds could not be measured in this work such as caffeine (saturated signal) or trigonelline (insufficiently retained on the chromatographic column). A combination of several chromatographic methods would be needed for a more comprehensive analysis of coffee compounds. Secondly, the number of annotated compounds was limited by the limited availability of chemical standards or the lack of high-quality mass spectra. Further analytical work would be needed for more comprehensive coverage of coffee compounds. In future studies, it will be of particular importance to develop targeted methods of analysis to measure in human biospecimens the coffee compounds specific to the different types of coffee brews as described here, but also to measure all major bioactive compounds known in coffee that may explain the effects of coffee intake on health. 

## 4. Materials and Methods

### 4.1. Standards and Reagents

UPLC-grade acetonitrile and formic acid were supplied by Sigma-Aldrich (France) and LCMS grade water was generated from a Milli-Q integral water purification system. Standards to aid compound identification were obtained as follows: N-methyl-2-pyrrolecarboxaldehyde, 5-caffeoylquinic acid, were purchased from Sigma, France; cyclo(leucyl-prolyl) and cyclo(prolyl-valyl) were purchased from Bachem, Switzerland; 3-caffeoylquinic acid and 3,5-dicaffeoylquinic acid were purchased from Extrasynthese, France; and cyclo(phenylalanyl-prolyl) was purchased from Chem-Impex, Ilinois, US. Standards of theophylline, theobromine, paraxanthine, caffeine, and guanosine were obtained as part of the Mass Spectrometry Metabolite Library (MSMLS) kit of standards by IROA Technologies (Bolton, MA, USA).

### 4.2. Purchase and Brew of Coffee Samples

A total of 76 unique coffee samples representing a diverse range of coffee brews were brewed or purchased in the Rockville, Maryland area, USA (Appendix A). We targeted brands that accounted for a majority of the market share to provide the best coverage of coffee exposure. For the ground coffees, we brewed all selections using a drip machine since this is the most common form of preparation in the US. Then, for comparison, we brewed ground beans from the same source (i.e., the bag) using the French Press, percolator, and cold brew methods, balancing these between 100% Arabica or blend of Arabica and Robusta, light, medium, or dark roast, and caffeinated and decaffeinated varieties. With limited resources for analysis and countless potential combinations of coffee types and brew methods, we focused on the most commonly consumed coffee types and brew methods. We tried to balance the number of samples in the compared groups to prevent factors other than the one being considered from impacting the results. Coffee brews prepared with Robusta bean exclusively are not commonly consumed in the US and were therefore not included in this study. Robusta beans are thus brewed as blends of Arabica and Robusta beans in proportions that are usually not disclosed by the food manufactures.

Ground coffee samples were brewed with a BUNN® drip coffee maker with a paper filter using 50 g of ground coffee per 1893 mL of tap water (i.e., 5 g ground coffee per 177 mL cup of tap water). The drip coffee machine heated the water to approximately 200 °F (93 °C). French Press coffees were brewed using 23 g of ground coffee per 798 mL of tap water. Water was on average 190 °F (88 °C) upon pouring, and French Press brews were left to steep at room temperature for 3 minutes. Percolated coffee was brewed using a Presto® Coffee Maker with a paper filter as described by the manufacturer using 40 g of ground coffee per 1183 mL of tap water. The Presto® coffee maker heated the water to approximately 200 °F (93 °C). Cold brew coffee concentrates were brewed using 24 g of ground coffee per 89 mL of tap water and were left in the refrigerator overnight. After 19 h, cold brew coffee concentrates were filtered using a paper filter and were then diluted using a 1:3 ratio of concentrate to tap water (i.e., 17 g ground coffee per 177 mL of tap water). 

K-cups were brewed using pods made for Keurig® coffee machines, each containing 10 g of coffee grounds and brewed with 177 mL tap water heated to approximately 192 °F (89 °C). Three types of espresso were brewed using a Nespresso® machine with pods containing 6 g of ground coffee and approximately 74 mL of water. Twelve unique types of instant coffee were prepared from single-serving sachets containing 2 g of powdered coffee and 177 mL tap water heated to approximately 183 °F (84 °C). Espresso brews were purchased from national chains serving coffee. Turkish and Greek boiled coffees were brewed per package instructions with 6 g of coffee grounds per 59 mL of tap water, which was heated to approximately 190 °F (88 °C) prior to pouring.

All coffees were brewed or purchased in triplicate. The triplicate preparations were combined to make a single pooled sample for each coffee type to limit the total number of analyses. Coffee samples were cooled and stored in a refrigerator, and two aliquots of each pooled sample were transferred to a −80 °F freezer within 6 hours of brewing. Two coffee samples were aliquoted 13 times each and used as blinded QCs for the monitoring of analytical repeatability. QC1 consisted of replicate samples of a medium roast, decaffeinated, 100% Arabica coffee, and QC2 of replicate samples of a dark roast, caffeinated, Arabica/Robusta blend coffee. 

### 4.3. Untargeted Chemical Profiling of Coffee Samples Using Liquid Chromatography-Mass Spectrometry

Coffee samples were thawed and agitated, and an aliquot of each was placed in Eppendorf tubes. Samples were centrifuged at 120× *g* for 10 min to separate each into a particulate pellet, with the clear aqueous portion and any fat from the coffee resting on top. Any fatty supernatant (in three instant coffee samples containing powdered milk) was first removed using a micro-spatula and a portion of the aqueous fraction was then taken, diluted 50-fold in water, and transferred to glass vials for instrumental analysis in a randomized order. An equal mixture of all coffee samples was also made and used as a pooled quality control (QC) sample. This QC was placed in the injection sequence after every 10 study samples, giving a total of 18 replicates. In addition, six aqueous methanol blanks were placed in the injection sequence after every 20 study samples, giving a total of 6 replicates, and blinded QC samples (2 × 13 samples) were randomly distributed across the sample batch. With column conditioning QCs, a total of 200 samples were thus injected into the instrument, requiring three 96-well injection plates.

Coffee brews were analyzed directly using liquid chromatography coupled to a time-of-flight mass spectrometer (LC-MS). The instrument used was an Agilent 1290 Binary LC system coupled to an Agilent 6550 quadrupole time-of-flight (QTOF) mass spectrometer with jet steam electrospray ionization source. Samples (2 μL) were injected onto a reversed-phase C18 column (ACQUITY UPLC HSS T3 2.1 × 100 mm, 1.8 μm, Waters) maintained at 45 °C. The mobile phase used was a mixture of ultrapure water and LC-MS grade methanol. A linear gradient made of ultrapure water (eluant A) and LC-MS grade methanol (eluant B) was used for elution: 0–100% B from 0–6 min, 100% B from 6–10.5 min, and 5% B for the remainder of the 13-min run. The mass spectrometer was operated in positive ionization mode and ions were detected across a mass range of 50–1000 daltons. Agilent Mass Hunter software and a recursive two-stage extraction pass were used to align and extract mass spectral data. First, all possible molecular features were extracted from every sample, and masses and retention times aligned according to tolerance windows of 15 ppm and 0.1 min, respectively. The resulting list was then used as a target for a second recursive extraction, which enabled features closer to the noise floor to be picked up. The result of this molecular feature extraction was a table of 3670 spectral features aligned across samples by mass to charge ratio (*m*/*z*) and retention time with accompanying relative intensities (peak areas). 

### 4.4. Mining of Untargeted Metabolomic Data for Known Coffee Compounds

Untargeted profile data acquired by LC-MS were mined for known coffee components. Firstly, chemical formulae and exact masses of as many known coffee components as possible were collected from online resources such as the Dictionary of Food Ingredients, Duke’s Phytochemical and Ethnobotanical database (www.phytochem.nal.usda.gov) and FooDB (www.foodb.ca). Data was collected for around 500 organic compounds (348 formulae). All formulae were then searched for directly in the raw data files of pooled QC samples using the Agilent Quantitative Analysis Find by Formula algorithm. This algorithm sums signals from different ion species to generate a peak for each formula. Peaks were found corresponding to 115 formulae matching those of the coffee compounds. In some cases, multiple peaks at different retention times were retained for the same formula.

Subsequently, 88 groups of peaks potentially corresponding to known coffee compounds remained and were checked for acceptable peak shapes and consistent retention times. An Agilent Personal Compound Database and Library (PCDL), populated using these masses and retention times, was then used as the target for further extraction using the Find by Formula algorithm on the 76 coffee samples. The resulting groups of peaks were carefully screened for inclusion. A total of 24 of these putative coffee compounds were then excluded due to detector saturation (caffeine only), large peaks in the injected blanks, missing peaks in >25% samples, or high coefficient of variation (>30%) in pooled or blinded QCs. For the remaining 64 peaks (Appendix A), mass spectral data were mined to confirm the initial hypotheses or, if disproved, generate alternative hypotheses. Annotations were thus made taking into account inter-compound correlations (Appendix A), and each attributed a level of confidence for annotation [23]. Level 1 corresponding to confirmed structure, based on comparison of retention time and MS/MS match with those of an authentic chemical standard. For level 2, no standard was available or analyzed, and probable identification was based on physicochemical properties, isotope pattern, and MS/MS spectra. For level 3, the tentative structure was based on comparison with a structurally related compound in the same chemical class and characteristic fragments in MS/MS spectra. For level 4, the tentative annotation was based on the unequivocal molecular formula. Level 5 correspond to unknown compounds. When no chemical standard was available, experimental MS/MS spectra were compared to library spectra in databases such as HMDB [38] and Metlin [39]. If MS/MS data were not available (for example, where the intensity of the [M + H]^+^ ion was low), retention time only was used. If identities could not be confirmed by retention time or MS/MS spectra matching, the assigned identity was regarded as hypothetical (level 4). Technical reproducibility of the semi-quantitative measurements of peak areas is shown for annotated compounds in Appendix A for the pooled QC samples and blinded duplicate QC samples.

### 4.5. Statistical Analysis

The untargeted feature table was first filtered to remove any features present in one or two samples only. All other features were retained. Multivariate data were log_2_ transformed and Pareto scaled, and constituent profiles were explored by performing a principal component analysis (PCA) on multivariate data for all unique brews (*n* = 76). The variability in the untargeted profile explained by each factor was then determined using principal component partial R-square (PC-PR2) multivariable regression [40]. The covariates included in the models were the brew method, bean type, roast, caffeine type, injection order, and injection plate. 

The effect of the most influential brew parameters on the profiles of 18 annotated compounds was then investigated using PCA for all 76 brews. Data were first normalized to total intensity (sum of all detected compounds; Appendix A) to remove the effect of differing brew strengths from the analysis [41] and were log-transformed. Loadings were calculated to determine which compounds or groups of compounds contributed most to the largest principal components (PC) of variation. An unpaired Student’s *t*-test was used to assess differences between different coffee subgroups (dark vs. medium roast, Arabica beans vs. blend, instant coffee vs other brew methods, regular coffee vs. decaffeinated coffee) for the 18 identified metabolites. Bonferroni-corrected p-values below 0.05 were considered significant. The PCA and PC-PR2 models were fit using R open-source statistical software. All analyses were performed using R software version 3.4.4 [42].

## 5. Conclusions

There is a high demand for better understanding of the health benefits or risks associated with the consumption of coffee, one of the most widely consumed beverages in the world. The coffee composition is highly variable, depending on coffee brew method, bean variety, level of roast and decaffeination. However, it is still unclear how these parameters affect coffee bioactivity and associations with disease. The current study is the first to compare the chemical profiles of the main types of coffee consumed in the USA and to identify factors that contribute to this variation. It is hoped that the results presented here will contribute to better estimations of coffee intake in epidemiological studies and to clarify the relationship with disease.

## Figures and Tables

**Figure 1 metabolites-09-00017-f001:**
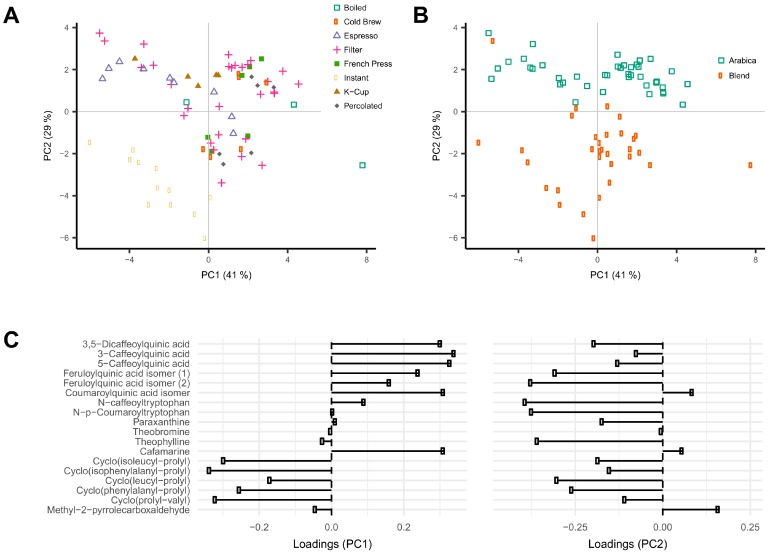
Principal component analysis (PCA) of constituent profiles of the 18 annotated coffee compounds measured in 76 coffee brew samples. Data were normalized to the sum of the intensities of all compounds. Scores and loadings on principle components PC1 and PC2 are shown. (**A**) Color-coding according to the type of coffee brew; (**B**) color-coding according to the type of coffee beans; (**C**) loading plots.

**Figure 2 metabolites-09-00017-f002:**
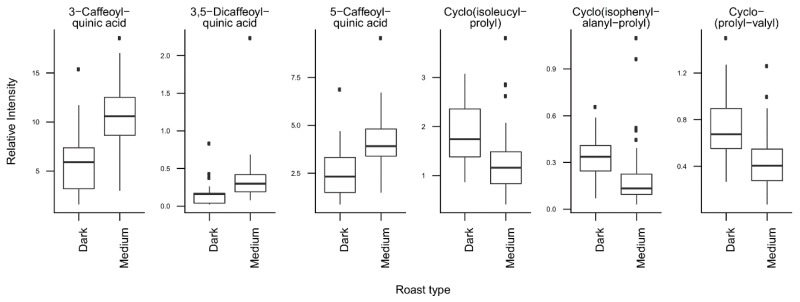
Relative concentrations of the main discriminant metabolites in dark roasted (*n* = 17) and medium roasted (*n* = 47) coffee brew samples.

**Figure 3 metabolites-09-00017-f003:**
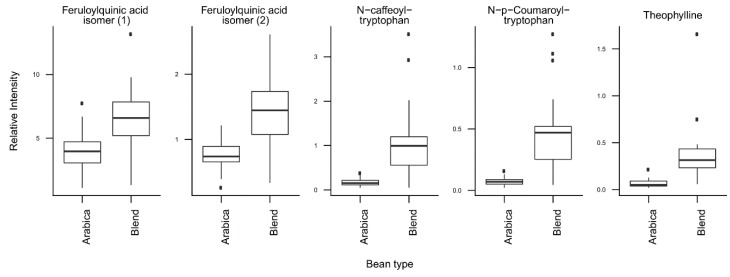
Relative concentrations of the main discriminant metabolites in coffee brews prepared with Arabica beans (*n* = 41) or blends of Arabica and Robusta beans (*n* = 32).

**Figure 4 metabolites-09-00017-f004:**
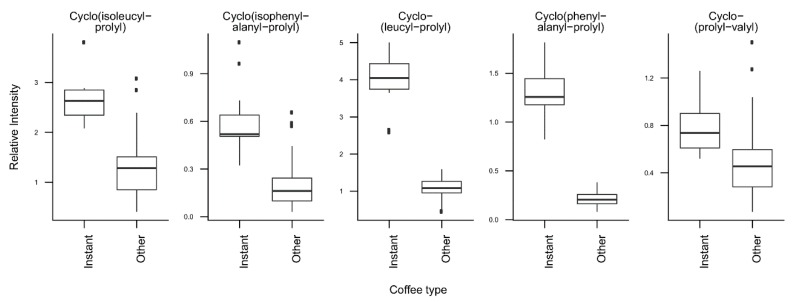
Relative concentrations of the main discriminant metabolites in instant coffees (*n* = 12) and all other coffee brew samples (*n* = 64).

**Figure 5 metabolites-09-00017-f005:**
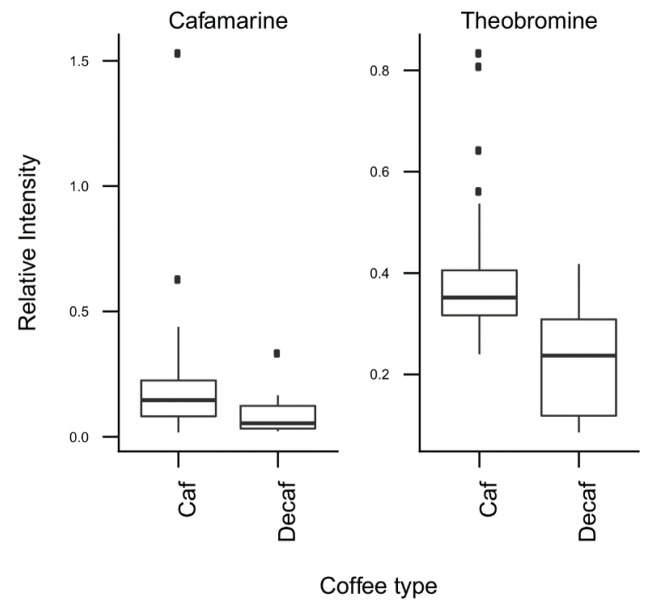
Relative concentrations of the main discriminant metabolites in decaffeinated (*n* = 16) and caffeinated (*n* = 60) coffee brews.

**Table 1 metabolites-09-00017-t001:** Coffee compounds annotated in 76 coffee brews.

Compound	Class	Formula	Observed *m*/*z*	Ion	Mass Difference (Delta ppm)	Retention Time (min)	Annotation Level ^1^
3,5-Dicaffeoylquinic acid	Phenolic acid ester	C_25_H_24_O_12_	517.1347	[M + H]^+^	0.63	3.85	1
3-Caffeoylquinic acid	Phenolic acid ester	C_16_H_18_O_9_	355.1030	[M + H]^+^	0.62	2.94	1
5-Caffeoylquinic acid	Phenolic acid ester	C_16_H_18_O_9_	355.1030	[M + H]^+^	0.90	2.46	1
Feruloylquinic acid isomer (i)	Phenolic acid ester	C_17_H_20_O_9_	369.1186	[M + H]^+^	1.27	3.47	3
Feruloylquinic acid isomer (ii)	Phenolic acid ester	C_17_H_20_O_9_	369.1186	[M + H]^+^	1.27	2.99	3
Coumaroylquinic acid isomer	Phenolic acid ester	C_16_H_18_O_8_	339.1081	[M + H]^+^	1.97	3.35	3
N-caffeoyltryptophan	Phenolic acid amide	C_20_H_18_N_2_O_5_	367.1295	[M + H]^+^	1.99	4.48	2
N-*p*-Coumaroyltryptophan	Phenolic acid amide	C_20_H_18_N_2_O_4_	351.1346	[M + H]^+^	2.41	4.71	2
Paraxanthine	Methylxanthine	C_7_H_8_N_4_O_2_	181.0726	[M + H]^+^	10.41	2.69	1
Theobromine	Methylxanthine	C_7_H_8_N_4_O_2_	181.0726	[M + H]^+^	3.75	2.37	1
Theophylline	Methylxanthine	C_7_H_8_N_4_O_2_	181.0726	[M + H]^+^	4.30	2.81	1
Cafamarine	Terpene glycoside	C_26_H_36_O_10_	509.2387	[M + H]^+^	0.50	4.05	3
Cyclo(isoleucyl-prolyl)	Diketopiperazine	C_11_H_18_N_2_O_2_	211.1447	[M + H]^+^	3.67	3.76	2
Cyclo(isophenylalanyl-prolyl)	Diketopiperazine	C_14_H_16_N_2_O_2_	245.1291	[M + H]^+^	5.01	3.94	2
Cyclo(leucyl-prolyl)	Diketopiperazine	C_11_H_18_N_2_O_2_	211.1447	[M + H]^+^	3.67	3.87	1
Cyclo(phenylalanyl-prolyl)	Diketopiperazine	C_14_H_16_N_2_O_2_	245.1291	[M + H]^+^	3.78	4.05	1
Cyclo(prolyl-valyl)	Diketopiperazine	C_10_H_16_N_2_O_2_	197.1291	[M + H]^+^	3.17	3.08	1
Methyl-2-pyrrolecarboxaldehyde	Heteroaromatic compound	C_6_H_7_NO	110.0607	[M + H]^+^	4.00	3.62	1

^1^ Confidence level for identification as defined by [23].

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
