# Peer review of "A Metabolomic Study of the Variability of the Chemical Composition of Commonly Consumed Coffee Brews"

_metabolites, 2019, doi:10.3390/metabo9010017_

Reviewer 1 Report

Notes for the Authors.

This is an excellent manuscript, and should certainly be published in the journal without significant changes.

A few points the authors might want to consider:

1.      The samples include brews prepared from 100 % arabica beans, and samples prepared from arabica – robusta blends, but no samples prepared 100 % robusta beans. If 100 % robusta samples had been included in the sample set, this should have resulted in greater statistical discrimination of those metabolites referred to in Figure 3, and to an even better resolution of PCA scores subpopulation clusters shown in Figure 1B. The reviewer does not mean to suggest here that the subpopulation resolution in Figure 1B is not convincing, because it certainly is, but simply that an even better resolution would be obtained if 100 % robusta samples were included in the sample set. However, the sample population examined in this study specifically targets coffee brews that are actually consumed by the coffee drinking population in the United States. It seems possible that no 100 % robusta brews are actual drunk in United States. Still if this is in fact the case the authors might want to include a sentence clarifying the situation.

2.      Do the authors know the percentage by weight of the arabica and robusta beans present in the blend samples? Does this percentage vary in commercially available blended coffee brews in the United States? In this study was this percentage constant over the sample set? If this information is available, the authors might want to include it in a brief sentence or two in the manuscript.

3.      In describing the sample set on line 213 the authors refer to three roast levels, “light”, “medium” and “dark”. However in Figure 3 only data from the “medium roast” and “dark roast” subpopulations are presented. What happened to the light roast samples?  The authors might want to include a sentence clarifying the fate of the light roast samples.

The manuscript is very well written, very clear as to intended meaning, and very readable through-out. Two minor points:

4.      In line 22 the authors refer to “41 % variability”. The reviewer believes that this should be “41 % variance”.

5.      Through-out the manuscript the authors refer to the sample population in this study as derived from 74 coffee brew samples. In the abstract on line 15, and again in the section describing the Supplementary Materials on line 317, the authors refer to 76 coffee brew samples. Please clarify. Is this difference due to the two samples that were separated out from the population to be used for analytical method quality control?

Author Response

Reply included in the attached Word file.

Reviewer 2 Report

The manuscript addresses an important topic, shedding light on the chemical composition of different coffee blends, roastings and preparations. This knowledge is important in order to link chemical compositions to biological effects of different coffees. 

My main concern with this manuscript is the rather limited number of chemical compounds and the bias in elucidated chemical structures. From the untargeted analysis, where 3670 features were detected, only 18 compounds could be identified – with different levels of certainty. As such, important (bioactive) compounds of coffee are not included in this study, such as furans, trigonelline, caffeine, lipids, carcinogens such as acrylamide, and many others. Some of which are also discussed to have adverse effects on health. 

The results are mainly based on PCA analysis of 18 compounds. I suggest to include an PCA analysis based on coffee „metadata“ such as: caffeination, roast level, brew method, coffee variety etc This should help to get an unbiased overview of how these factors influence the chemical composition. 

Specific comments: 

-      line 20: revise sentence ...to be...

-      Introduction: a comment on potential adverse effects should be added. 

-      Methods: There is a high chance of bias from total intensity normalization, when few compounds have very different concentrations in the spectrum. Have you considered other normalization techniques than total intensity?  eg. to dry weight (to account for dilution) or probabalistic quotient normalization. 

-      Metabolite identification: For levels of identification, I suggest to refer to the publication from Sumner et al Metabolomics 2007, doi:10.1007/s11306-007-0082-2

-      The discussion is rather a repetition and summary of the results. I suggest to revise the discussion section, and also include: provide the a good review of to date known compounds in coffee and show what fraction of the to date known compounds in coffee are covered or added in this manuscript; compare your analytical method to other methodological possibilities and their advantages/disadvantages; mention the possibility of using urine biomarkers of coffee in order to differenciate coffee consumptions (e.g. metabolites of furans and trigonelline, as published elsewhere)

Reviewer 3 Report

The topic of the paper is of great interest, as demonstrated by the literature in this field.

My main concern is the analytical method used by the authors to analyze through an agnostic untargeted metabolomic approach a wide range of coffee brews” that allowed to consider a reduced panel coffee metabolites, excluding some compounds with significant impact on human health, as trigonelline and caffeine. To obtain an exhaustive work, these compounds should be evaluated, setting the appropriate experimental conditions to reach this goal. Moreover, the adverse effects of some important coffee metabolites (caffeine) should be considered in this work.

The aim of the paper is clearly stated, but it is not clear if it was fulfilled or not. The authors should explain more extensively the implications of their results, especially concerning the influence of the brewing method on the chemical composition of coffee, with particular focus on the bioactive metabolites. In this perspective, conclusions paragraph could be more incisive.

Minor comments:

1.    Line 40. Previous literature on the analysis of bioactive compounds in different coffee brews should be cited (Niseteo, T., Komes, D., Belščak-Cvitanović, A., Horžić, D., & Budeč, M. (2012). Bioactive composition and antioxidant potential of different commonly consumed coffee brews affected by their preparation technique and milk addition. Food Chemistry, 134(4), 1870–1877; C. Ciaramelli, A. Palmioli, C. Airoldi. Coffee variety, origin and extraction procedure: Implications for coffee beneficial effects on human health. Food Chemistry 278 (2019) 47–55).

2.    Line 67. Title of paragraph 2.2 could be more informative.

3.    Line 81. Please clarify the meaning of “Association of coffee compound profiles with different types of coffee brews.”.

4.    Line 100. “Caffeinated and decaffeinated coffee brews could be differentiated along PC3 (accounting for 7.7% of the variability), a result consistent with high loadings of paraxanthine and theobromine on this PC (Supplementary figure S3)”. Please move this sentence to main text and explain why in fig S3 A the PC2 and PC3 score plots of coffee brews are reported: do they contain additional information respect to PC1 and PC2?

5.    Figures 2, 3, 4 and 5 can be rationalised and merged. The quality of the pictures should be improved.

6.    Line 172. Authors cite a review of 2014 regarding the impact of coffee consumption on health (Ref. 20). There is very recent literature on the properties of coffee and CGAs on human health that should be mentioned, including prevention in neurodegenerative diseases and cancer (a) F. Panza, V. Solfrizzi, M.R. Barulli, C. Bonfiglio, V. Guerra, A. Osella, D. Seripa, C. Sabbà, A. Pilotto, G. Logroscino. J Nutr Health Aging,19, 2015; b) C. Ciaramelli, A. Palmioli, A. De Luigi, L. Colombo, G. Sala, C. Riva, C. P. Zoia, M. Salmona, C. Airoldi. Food Chemistry, 2018, 252, 171–180; c) A. Palmioli, C. Ciaramelli, R. Tisi, M. Spinelli, G. De Sanctis, Elena Sacco, C. Airoldi. Chem. Asian J. 2017, 12, 2457 – 2466; d) N. Liang, D. D. Kitts, Nutrients 2016, 8, 16)

7.    Line 215. “The triplicate preparations were combined to make a single pooled sample for each coffee type.” The authors should explain the reason of pooling the triplicate samples and not analysing them in three different experiments.

8.    Line 220. “Coffee selection and preparation are described in further detail elsewhere (Loftfield et al, in preparation).” Please provide the details in this paper, not in a manuscript in preparation.

Regarding the coffee selection, an explanation of the rationale behind the selection of coffee samples should be added. Could the authors also clarify the reason of applying a different set of brewing methods for the different brands? Consequently, different numbers of samples are produced for each type of coffee brew: could this have any impact on the results?

9.    Line 224. “Samples were centrifuged at 120g.” Please correct 120g.

10.  Line 238 - 239. Please correct 2 uL and 45oC

Author Response

See attached file

Round  2

Reviewer 3 Report

I am pleased that the authors performed a good job in the acceptance of the suggestions of corrections. I am convinced that the quality of the manuscript has improved. Therefore, I suggest to accept the paper as it is now.

Only 2 minor comments on the new experimental paragraphs:

-       Substitute fluid ounces (fl oz) with mL

-       Temperature unit: correct with a space between the number and °C